# Cladolosides of Groups S and T: Triterpene Glycosides from the Sea Cucumber *Cladolabes schmeltzii* with Unique Sulfation; Human Breast Cancer Cytotoxicity and QSAR

**DOI:** 10.3390/md23070265

**Published:** 2025-06-25

**Authors:** Alexandra S. Silchenko, Elena A. Zelepuga, Ekaterina A. Chingizova, Ekaterina S. Menchinskaya, Kseniya M. Tabakmakher, Anatoly I. Kalinovsky, Sergey A. Avilov, Roman S. Popov, Pavel S. Dmitrenok, Vladimir I. Kalinin

**Affiliations:** G.B. Elyakov Pacific Institute of Bioorganic Chemistry, Far Eastern Branch of the Russian Academy of Sciences, Pr. 100-letya Vladivostoka 159, 690022 Vladivostok, Russia; zel@piboc.dvo.ru (E.A.Z.); chingizova_ea@piboc.dvo.ru (E.A.C.); ekaterinamenchinskaya@gmail.com (E.S.M.); tabakmakher_km@piboc.dvo.ru (K.M.T.); kaaniv@piboc.dvo.ru (A.I.K.); avilov_sa@piboc.dvo.ru (S.A.A.); popov_rs@piboc.dvo.ru (R.S.P.); paveldmt@piboc.dvo.ru (P.S.D.)

**Keywords:** *Cladolabes schmeltzii*, Dendrochirotida, Sclerodactylidae, triterpene glycosides, sea cucumber, hemolytic and cytotoxic activity, human breast cancer, QSAR

## Abstract

Four new minor monosulfated triterpene penta- and hexaosides, cladolosides S (**1**), S_1_ (**2**), T (**3**), and T_1_ (**4**), were isolated from the Vietnamese sea cucumber *Cladolabes schmeltzii* (Sclerodactylidae, Dendrochirotida). The structures of the compounds were established based on extensive analysis of 1D and 2D NMR spectra as well as HR-ESI-MS data. Cladodosides S (**1**), S_1_ (**2**) and T (**3**), T_1_ (**4**) are two pairs of dehydrogenated/hydrogenated compounds that share identical carbohydrate chains. The oligosaccharide chain of cladolosides of the group S is new for the sea cucumber glycosides due to the presence of xylose residue attached to C-4 Xyl1 in combination with a sulfate group at C-6 MeGlc4. The oligosaccharide moiety of cladolosides of the group T is unique because of the position of the sulfate group at C-3 of the terminal sugar residue instead of the 3*-O*-Me group. This suggests that the enzymatic processes of sulfation and *O*-methylation that occur during the biosynthesis of glycosides can compete with each other. This can presumably occur due to the high level of expression or activity of the enzymes that biosynthesize glycosides. The mosaicism of glycoside biosynthesis (time shifting or dropping out of some biosynthetic stages) may indicate a lack of compartmentalization inside the cells of organism producers, leading to a certain degree of randomness in enzymatic reactions; however, this also offers the advantage of providing chemical diversity of the glycosides. Analysis of the hemolytic activity of a series of 26 glycosides from *C. schmeltzii* revealed some patterns of structure–activity relationships: the presence or absence of 3*-O*-methyl groups has no significant impact, hexaosides, which are the final products of biosynthesis and predominant compounds of the glycosidic fraction of *C. schmeltzii*, are more active than their precursors, pentaosides, and the minor tetraosides, cladolosides of the group A, are weak membranolytics and therefore are not synthesized in large quantities. Two glycosides from *C. schmeltzii*, cladolosides D (**18**) and H_1_ (**26**), display selectivity of cytotoxic action toward triple-negative breast cancer cells MDA-MB-231, while remaining non-toxic in relation to normal mammary cells MCF-10A. Quantitative structure–activity relationships (QSAR) were calculated based on the correlational analysis of the physicochemical properties and structural features of the glycosides and their hemolytic and cytotoxic activities against healthy MCF-10A cells and cancer MCF-7 and MDA-MB-231 cell lines. QSAR highlighted the complexity of the relationships as the cumulative effect of many minor contributions from individual descriptors can have a significant impact. Furthermore, many structural elements were found to have different effects on the activity of the glycosides against different cell lines. The opposing effects were especially pronounced in relation to hormone-dependent breast cancer cells MCF-7 and triple-negative MDA-MB-231 cells.

## 1. Introduction

Sea cucumbers (Holothuroidea, Echinodermata) are marine invertebrates that have become a focus of extensive chemical investigations because they produce pharmacologically relevant triterpene glycosides, making them a valuable source of novel bioactive compounds. Despite the long history of investigations of the triterpene glycosides of different species of sea cucumbers, they continue to attract the attention of scientists due to their intriguing chemical diversity and various biological activities [1,2]. Some of the recent investigations of the glycosidic composition of sea cucumbers are based on mass-spectrometric metabolomic profiling [3,4,5]. However, this approach should be used in conjunction with the isolation and structure elucidation of individual native glycosides, since it does not allow the unambiguous determination of the spatial structure of substances and the distinction of isomers, especially given that modern techniques and technologies allow the separation of complex mixtures of polar native glycosides and isolation of minor compounds [6]. Nevertheless, metabolomic studies of sea cucumbers provide useful data on the chemical diversity of glycosides and their content inside the different internal parts of animals, as well as ecological functions [7,8,9,10].

In addition, glycosides have promising and diverse biological activities [5,11,12,13,14]. Recently, a selective cytotoxic effect of glycosides against triple-negative breast cancer cells (MDA-MB-231) was discovered [15,16,17], suggesting their potential use in targeted therapy of breast cancer.

The main patterns of structure–activity relationships (SAR) of these metabolites were deduced earlier [18,19], but the application of correlational analysis for quantitative structure–activity relationship (QSAR) calculations [16,20] based on a broadened combinatorial library will provide further opportunities to predict the bioactivity of glycosides.

Previously, 40 glycosides (cladolosides of groups A–R (**5**–**44**, Appendix A)) characterized by extreme variability of carbohydrate chains were isolated from the sea cucumber *Cladolabes schmeltzii* (Sclerodactylidae, Dendrochirotida) [21,22,23,24,25,26]. The anticancer action of 14 cladolosides against colorectal adenocarcinoma HT-29 was tested, revealing varying effects on the inhibition of colony formation. Some glycosides demonstrated a synergistic effect with radioactive irradiation, increasing the inhibitory effect of radiation against cancer cells [26].

The minor glycosides, cladolosides S (**1**), S_1_ (**2**), T (**3**), and T_1_ (**4**), were isolated in the final study on the glycosidic composition of the sea cucumber *C. schmeltzii*. Their structures were established by analyzing ^1^H, ^13^C NMR, 1D TOCSY, and 2D NMR (^1^H,^1^H COSY, HMBC, HSQC, ROESY), and HR-ESI mass spectra. All original spectra are provided in Appendix A. The hemolytic activity against human erythrocytes and the cytotoxicity against human breast cancer cell lines MCF-7, T-47D, and triple-negative MDA-MB-231, as well as the non-tumor mammary epithelial cell line MCF-10A, for the series of 26 cladolosides, including isolated earlier and novel compounds, were tested. QSAR analysis was conducted for the same series of compounds.

## 2. Results and Discussion

### 2.1. Structure Elucidation of Glycosides

Specimens of *C. schmeltzii* were collected from the Nha Trang Gulf in the East Sea of Vietnam. Glycosides were isolated from the concentrated ethanol extract using a previously described methodology [21]. Silica gel column chromatography with different ratios of solvents CHCl_3_/EtOH/H_2_O as the mobile phase gave several main fractions with varying polarity, as well as two minor fractions weighing 20 and 14 mg. The isolation of individual compounds was achieved by subsequent HPLC of the latter fractions on reversed-phase and silica-based columns to give four glycosides **1**–**4** (Figure 1).

Absolute configurations of all monosaccharide residues (xylose, quinovose, glucose, and 3*-O*-methylglucose) comprising the sugar chains of the glycosides of *C. schmeltzii* were assigned earlier as D using acid hydrolysis of the total glycosidic fraction with TFA, followed by alcoholysis with (R)-(−)-2-octanol and subsequent acetylation and GLC analysis (see Appendix A for methodology and GLC data of modified monosaccharides) [23].

The molecular formula of cladoloside S (**1**) was determined to be C_62_H_95_O_31_SNa from the [M_Na_ − Na]^−^ ion peak at *m*/*z* 1367.5580 (calc. 1367.5584) and [M_Na_ − Na − H]^2−^ ion peak at m/z 683.2766 (calc. 683.2755) in (−)HR-ESI-MS (Appendix A). As deduced from extensive analysis of the NMR spectra, the aglycone of **1** was typical for the major part of the glycosides from *C. schmeltzii* [21,23,24,25,26,27,28,29] and contained 18(20)-lactone, 9(11)- and 25(26)-double bonds, and 16*S*- and 22*R-O*-acetic groups (Table 1, Appendix A).

The analysis of the ^1^H, ^13^C NMR, and HSQC spectra of the carbohydrate moiety of cladoloside S (**1**) (Table 2; Appendix A) revealed the presence of five doublets of the anomeric protons at *δ*_H_ 4.66–5.27 (*J* = 6.8–8.3 Hz) and the signals of the corresponding anomeric carbons at *δ*_C_ 102.2–104.8. These data indicate that the pentasaccharide chain with *β*-glycosidic bonds is inherent to glycoside **1**. Starting from the NOE-correlation of H-3 of the aglycone (*δ*_H_ 3.25 (H-3, dd, *J* = 12.0; 3.9 Hz) (typical position of sugar chain attachment) with the anomeric proton of the first sugar unit at *δ*_C_ 4.66 (H-1 Xyl1, d, *J* = 6.9 Hz), observed in the ROESY spectrum, followed by analysis of the ^1^H, ^1^H COSY, 1D TOCSY, and HSQC spectra of each monosaccharide residue, the structure of the carbohydrate moiety of **1** was established. From this analysis, it was concluded that the oligosaccharide moiety of **1** consisted of quinovose (as the second unit in the chain), three xylose units (as the first, third, and fifth units), and 3*-O*-methylglucose as the terminal unit. ROESY and HMBC correlations allowed the determination of the positions of glycosidic bonds: correlations were observed between the H-1 Xyl1 and H-3 (C-3) of the aglycone, H-1 Qui2 and H-2 (C-2) Xyl1, H-1 Xyl3 and H-4 (C-4) Qui2, H-1 MeGlc4 and H-3 (C-3) Xyl3, and H-1 Xyl5 and H-4 (C-4) Xyl1 (Table 2). The single sulfate group was attached to C-6 of 3*-O*-methylglucose, causing an *α*-shifting effect of its signal to *δ*_C_ 66.6, and a *β*-shifting effect of the signal of C-5 MeGlc4 to *δ*_C_ 76.3 [24].

The (−)HR-ESI-MS/MS of **1** (Appendix A) demonstrated the fragmentation of the [M_Na_ − Na]^−^ ion at *m*/*z* 1367.5 leading to the fragment ions appearance at *m*/*z* 1307.5 [M_Na_ − Na − CH_3_COOH]^−^, 1247.5 [M_Na_ − Na − 2CH_3_COOH]^−^, 1235.5 [M_Na_ − Na − Xyl]^−^, 1175.5 [M_Na_ − Na − Xyl − CH_3_COOH]^−^, 797.2 [M_Na_ − Na − Agl − H]^−^, 533.1 [M_Na_ − Na − Agl − 2Xyl]^−^, and 387.0 [M_Na_ − Na − Agl − 2Xyl − Qui]^−^, corroborating the structure of **1** elucidated by analysis of NMR spectra.

These data indicate that cladoloside S (**1**) is 3*β-O*-{6*-O*-sodium sulfate-3*-O*-methyl-*β*-D-glucopyranosyl-(1→3)-*β*-D-xylopyranosyl-(1→4)-*β*-D-quinovopyranosyl-(1→2)-[(1→4)-*β*-D-xylopyranosyl]-*β*-D-xylopyranosyl}-16*S*,22*R*-diacetoxyholosta-9,25-diene.

The molecular formula of cladoloside S_1_ (**2**) was determined to be C_62_H_97_O_31_SNa based on the [M_Na_ − Na]^−^ ion peak at *m*/*z* 1369.5735 (calc. 1369.5740), and the [M_Na_ − Na − H]^2−^ ion peak at *m*/*z* 684.2849 (calc. 684.2834) in (−)HR-ESI-MS (Appendix A). The aglycone of cladoloside S_1_ (**2**) differed from that of **1** only in the absence of a terminal double bond in its side chain. Hence, the signals of polycyclic systems in the ^13^C NMR spectra of **1** and **2** were almost identical. In the spectrum of **2** (Table 3, Appendix A), the signals of geminal methyl groups C-26 and C-27 and tertiary C-25 were observed at *δ*_C_ 22.5, 22.8, and 28.1, respectively. This aglycone is also characteristic of the glycosides of *C. schmeltzii* [21,23,25,26,27].

The carbohydrate chain of cladoloside S_1_ (**2**) was identical to that of cladoloside S (**1**), as confirmed by its 2D NMR spectra (Appendix A).

The fragment ion peaks in the (−)HR-ESI-MS/MS of **2** (Appendix A) were observed at *m*/*z* 1309.5 [M_Na_ − Na − CH_3_COOH]^−^, 1249.5 [M_Na_ − Na − 2CH_3_COOH]^−^, 1177.5 [M_Na_ − Na − Xyl − CH_3_COOH]^−^, differing from the corresponding ion-peaks observed in the spectrum of cladoloside S (**1**) by 2 *amu* due to the presence of hydrogenated side chain. The series of fragment ion peaks: 797.2 [M_Na_ − Na − Agl − H]^−^, 533.1 [M_Na_ − Na − Agl − 2Xyl]^−^, and 387.0 [M_Na_ − Na − Agl − 2Xyl − Qui]^−^ are characterized by the same masses as in the spectrum of **1**, which corroborates the identity of carbohydrate chains of cladolosides of the group S.

Thus, cladoloside S_1_ (**2**) is 3*β-O*-{6*-O*-sodium sulfate-3*-O*-methyl-*β*-D-glucopyranosyl-(1→3)-*β*-D-xylopyranosyl-(1→4)-*β*-D-quinovopyranosyl-(1→2)-[(1→4)-*β*-D-xylopyranosyl]-*β*-D-xylopyranosyl}-16*S*,22*R*-diacetoxyholost-9-en.

Thus, cladodosides S (**1**) and S_1_ (**2**) are a pair of dehydrogenated/hydrogenated compounds sharing the identical carbohydrate chain, which is characteristic of *C. schmeltzii*. The sugar moiety of **1** and **2** is new for sea cucumber glycosides due to the presence of xylose attached to C-4 Xyl1, which is rather rare for this group of metabolites. Glycosides with a xylose residue that branches the oligosaccharide chain at C-4Xyl1 were first discovered in this species of sea cucumber. These are cladolosides of groups D (**18**–**20**), R (**44**) (with hexasaccharide chains) [21,26] and E (**21**, **22**) (with pentasaccharide chains) [23] (Appendix A). Cladolosides of the group E are desulfated derivatives of cladolosides of the group S (**1**, **2**) because the latter pair contains a sulfate group at C-6 MeGlc4.

The molecular formula of cladoloside T (**3**) was determined to be C_69_H_107_O_37_SNa from the [M_Na_ − Na]^−^ ion peak at *m*/*z* 1559.6233 (calc. 1559.6217) and [M_Na_ − Na − H]^2−^ ion peak at m/z 779.3096 (calc. 779.3072) in (−)HR-ESI-MS (Appendix A). The aglycone of **3** was identical to that of **1**, which was deduced from the identity of ^13^C NMR spectra of their aglycone parts (Appendix A).

In ^1^H and ^13^C NMR spectra of the carbohydrate moiety of cladoloside T (**3**) (Table 4; Appendix A) six doublets of the anomeric protons at *δ*_H_ 4.66–5.27 (*J* = 6.8–8.3 Hz) and the signals of the corresponding anomeric carbons at *δ*_C_ 102.2–104.8 were observed indicating the presence of hexasaccharide chain with *β*-glycosidic bonds. The same algorithm was applied for structure elucidation as described above. In the monosaccharide composition of **3** two xylose, one quinovose, two glucose, and one 3*-O*-methylglucose residues were found. The positions of the glycosidic bonds deduced from the ROESY and HMBC correlations were typical for all glycosides of holothuroids. Xylose and glucose residues were third and fifth units, correspondingly, in the chain of **3**. Analysis of the chemical shifts of the signals of the terminal residue attached to C-3 Glc5 showed that it was a 3*-O*-methylated glucose because the HMBC correlations *O*Me (*δ*_H_ 3.80, s)/C-3 MeGlc6 (*δ*_C_ 86.9) and H-3 MeGlc6 (*δ*_H_ 3.67 (t, *J*=9.1 Hz)/*O*Me (*δ*_C_ 60.7) were observed. Hence, the single *O*Me group was attached to the upper semi-chain, and the terminal (glucose) residue in the bottom semi-chain was non-methylated. However, the signal of C-3 Glc4 was deshielded to *δ*_C_ 84.2 as well as the signal of the corresponding proton (*δ*_H-3 Glc4_ 5.03, t, *J* = 8.3 Hz). Taking into consideration the presence of the sulfate group deduced from the mass spectra of **3**, as well as the absence of sulfation of the hydroxymethylene group (C-6 Glc4) due to its signal being observed at *δ*_C_ 61.7, the position of the sulfate group was assigned as C-3 Glc4 [24].

The (−)HR-ESI-MS/MS of **3** (Appendix A) demonstrated the fragmentation of the [M_Na_ − Na]^−^ ion at *m*/*z* 1559.6 that resulted in the fragment ions appearance at *m*/*z* 1499.6 [M_Na_ − Na − CH_3_COOH]^−^, 1439.6 [M_Na_ − Na − 2CH_3_COOH]^−^, 1383.6 [M_Na_ − Na − MeGlc]^−^, 1323.5 [M_Na_ − Na − MeGlc − CH_3_COOH]^−^, 1161.5 [M_Na_ − Na − CH_3_COOH − MeGlc − Glc]^−^, 1101.5 [M_Na_ − Na − 2CH_3_COOH − MeGlc − Glc]^−^, 1007.3 [M_Na_ − Na − Agl − H]^−^, 519.1 [M_Na_ − Na − Agl − MeGlc − Glc − Xyl]^−^, confirming the sequence of monosaccharide residues as well as aglycone structure of **3**.

These data indicate that cladoloside T (**3**) is 3*β-O*-{3*-O*-sodium sulfate-*β*-D-glucopyranosyl-(1→3)-*β*-D-xylopyranosyl-(1→4)-*β*-D-quinovopyranosyl-(1→2)-[3*-O*-methyl-*β*-D-glucopyranosyl-(1→3)-*β*-D-glucopyranosyl-(1→4)]-*β*-D-xylopyranosyl}-16*S*,22*R*-diacetoxyholosta-9,25-diene.

Cladoloside T_1_ (**4**) was determined to be a hydrogenated derivative of **3** from the molecular formula C_69_H_109_O_37_SNa, deduced from the [M_Na_ − Na]^−^ ion peak at *m*/*z* 1561.6377 (calc. 1561.6374) in (−)HR-ESI-MS (Appendix A). Therefore, the aglycone of **4** was the same as that of cladoloside S_1_ (**2**), which was confirmed by the coincidence of the ^13^C NMR spectra of its aglycone parts (Appendix A). The carbohydrate chain of cladoloside T_1_ (**4**) (Appendix A) was identical to that of cladoloside T (**3**); hence, close proximity of the signals in the ^13^C NMR spectra of their oligosaccharide moieties was observed.

The (−)HR-ESI-MS/MS spectrum of **4** (Appendix A) demonstrated fragmentation of the [M_Na_ − Na]^−^ ion at *m*/*z* 1561.6. Thus, the fragment ions were observed at *m*/*z* 1501.6 [M_Na_ − Na − CH_3_COOH]^−^, 1367.6 [M_Na_ − Na − MeGlc − H]^−^, 1325.5 [M_Na_ − Na − MeGlc − CH_3_COOH]^−^, 1163.5 [M_Na_ − Na − CH_3_COOH − MeGlc − Glc ]^−^, 1103.5 [M_Na_ − Na − 2CH_3_COOH − MeGlc − Glc]^−^, 1007.3 [M_Na_ − Na − Agl − H]^−^, 519.1 [M_Na_ − Na − Agl − MeGlc − Glc − Xyl]^−^, confirming the structure of **4**.

These data indicate that cladoloside T_1_ (**4**) is 3*β-O*-{3*-O*-sodium sulfate-*β*-D-glucopyranosyl-(1→3)-*β*-D-xylopyranosyl-(1→4)-*β*-D-quinovopyranosyl-(1→2)-[3*-O*-methyl-*β*-D-glucopyranosyl-(1→3)-*β*-D-glucopyranosyl-(1→4)]-*β*-D-xylopyranosyl}-16*S*,22*R*-diacetoxyholost-9-en.

Cladolosides T (**3**) and T_1_ (**4**) are another pair of compounds that differ in the presence or absence of a terminal double bond in their side chain and have identical to each other carbohydrate chains. The oligosaccharide moiety of cladolosides of the group T is unique because of the position of the sulfate group at C-3 of the terminal sugar residue. This suggests that the processes of sulfation and *O*-methylation occurring during the biosynthesis of glycosides can compete with each other.

Sulfation is generally considered the final stage of sugar moiety biosynthesis. In fact, cladolosides of the groups I, J, K, L (**28**–**33**) [25] (Appendix A), and S (**1**, **2**) are penta- and hexaosides with the sulphate groups attached to the terminal 3*-O*-methylated monosaccharide residues [25], while 3*-O*-methylation is a signal of termination, which stops further elongation of the biosynthesizing carbohydrate chain. However, the incorporation of 3*-O*-methyl groups into hexasaccharide chains does not occur simultaneously: first, it is attached to one semi-chain (upper or bottom), while another semi-chain could be subsequently elongated and then 3*-O*-methylated. The time shift (delay) of these stages in relation to each other, which is characteristic of the mosaic type of biosynthesis [6], leads to the formation of carbohydrate chains with 3*-O*-methylated or non-methylated terminal residues. This is well illustrated by the structures of the cladolosides of groups D (**18**–**20**) [21], P (**39**–**42**), Q (**43**), and R (**44**) [26,27] (Appendix A), and kurilosides of groups A, D, and E − glycosides from the sea cucumber *Thyonidium kurilensis* [28].

In highly polar glycosides, such as psolusosides—compounds found in the sea cucumber *Psolus fabricii* [29] and bearing up to four sulfate groups, two sulfate groups can bond to one monosaccharide residue. This broadened the number of known possible positions of introducing the sulfate group. As a result, they are not only attached to the C-6 of hexose residues, which is common for glycosides, but also to the C-2 and C-4 positions. However, sulfation did not compete with 3*-O*-methylation. Some glycosides of the sea cucumber *Paracaudina chilensis* contain a sulfate group at C-3 Glc3, which prevents further elongation of the carbohydrate chain during biosynthesis. This suggests the following assumption: sulfatase competes with glycosyltransferases. A similar situation was observed in the glycosides of *C. schmeltzii*, but another pair of enzymes rival each other: 3*-O*-methylase and sulfatase. This can presumably be explained by the high level of expression or activity of the enzymes involved in glycoside biosynthesis. Therefore, these enzymes can compete with each other. The sea cucumber species *C. shmeltzii*, *P. fabricii* [29], and *P. chilensis* [30], which are characterized by a great variety of carbohydrate parts of their glycosides, are good examples of this phenomenon.

Triterpene glycosides of the sea cucumbers are the products of the mosaic type of biosynthesis, when the parts of one molecule, aglycones and carbohydrate chains, are biosynthesized simultaneously and independently from each other [6]. The different stages of biosynthesis can occur at different times relative to each other, or even be skipped, resulting in significant structural diversity in glycosides. The mosaicism of glycoside biosynthesis may indicate a lack of compartmentalization inside the cells of organisms–producers that leads to a certain degree of randomness in enzymatic reactions. However, this offers the advantage of increasing the chemical diversity of the glycosides.

### 2.2. Biological Activity of the Glycosides

The hemolytic activity of 26 glycosides (4 novel and 22 known compounds, available in our collection) isolated from *C. schmeltzii* was tested against human erythrocytes (Table 5). Cladolosides T (**3**) and T_1_ (**4**) were among the most active compounds (ED_50_ 0.39 and 0.12 μM, respectively). This suggests that the sulfate group attached instead of the 3*-O*-Me group to the terminal sugar residue does not decrease membranolytic activity. Cladolosides S (**1**) and S_1_ (**2**) were only moderately active in relation to erythrocytes. Their activity is comparable to cladolosides I_2_ (**29**) and I_1_ (**28**) (Appendix A), which have the same aglycones and sulfate group positions. Compounds **28**, **29** differ from **1** and **2** in that they contain glucose as the fifth sugar in the chain instead of xylose. Thus, the sulfate group attached to C-6 of the terminal monosaccharide slightly reduces the glycosides’ membranolytic action, which aligns with earlier observations [20].

Surprisingly, none of the cladolosides of group A, having a linear tetrasaccharide chain and holostane aglycones, which commonly provide high activity, were effective hemolytics. Among the most active compounds were cladolosides of the group B (pentaosides with one glucose residue in upper semi-chain, ED_50_ 0.55 μM (cladoloside B_1_ (**11**)) and 0.31 μM (cladoloside B_2_ (**12**)) and group C (hexaosides with glucose and 3*-O*-methyl glucose in the upper semi-chain, ED_50_ 0.44 μM (cladoloside C (**13**)) and 0.15 μM (cladoloside C_1_ (**14**)) (Appendix A). There were no significant differences in activity between hydrogenated/dehydrogenated pairs of glycosides. Cladoloside D (**18**), hexaoside, having non-methylated terminal residue in the upper semi-chain was also highly active (ED_50_ 0.29 μM). Its 3*-O*-methylated analog, cladoloside G (**25**), is also highly hemolytic (ED_50_ 0.35 μM) (Appendix A). These facts indicate that the 3*-O*Me groups do not significantly influence the hemolytic activity of hexaosides.

Two pairs of glycosides, differing in the presence/absence of a 25(26)-double bond within each pair, demonstrated that hydrogenated compounds were more active than their dehydrogenated analogs. This pattern was observed for sulfated pentaosides, cladolosides of the group I (ED_50_ 1.72 μM (cladoloside I_1_ (**28**) and 4.24 μM (cladoloside I_2_ (**29**)), and also for sulfated hexaosides, cladolosides of the group K (ED_50_ 0.31 μM (cladoloside K_1_ (**31**) and 4.63 μM (cladoloside K_2_ (**32**)) (Appendix A).

Cladolosides L_1_ (**33**), M_1_ (**35**), and P_2_ (**41**) (Appendix A), having the same aglycones (holotoxinogenin, with 16-oxo group and 25(26)-double bond) and differing by monosaccharide composition of hexasaccharide chains, were all strong hemolytics (Table 5).

The weak activities of cladoloside C_3_ (**16**) (ED_50_ 12.10 μM) was caused by the presence of 16*-O*H group instead of common for the majority of glycosides 16*-O*Ac, of cladoloside A_3_ (**7**) (ED_50_ 7.58 μM) − by the absence of 22*-O*Ac, of cladoloside A_5_ (**9**) (ED_50_ 7.19 μM) − by the presence of 25-OH-group. The weakest effect from the series was demonstrated by cladoloside A_4_ (**8**) (Appendix A), with a 24(25)-double bond and 16-oxo-group instead of 16β*-O*Ac in the other cladolosides of group A, in combination with the tetrasaccharide chain. This is unusual because these features are not known to reduce the hemolytic activity of glycosides [20].

The cytotoxic activities of the same series of compounds were studied against three types of human breast cancer cells (MCF-7, T-47D, and triple-negative MDA-MB-231) as well as the non-tumor mammary epithelial cell line MCF-10A. Cisplatin and cucumarioside A_0_-1 [15] were used as positive controls (Table 5).

MDA-MB-231 cells were most susceptible to the cytotoxic action of the glycosides, whereas MCF-7 and T-47D cell lines were equally more viable. The most promising compounds were those that showed no toxicity to normal cells, MCF-10A, but were highly active against triple-negative cancer cells, MDA-MB-231. Cladolosides B_2_ (**12**), D (**18**), G (**25**), H_1_ (**26**), and I_1_ (**28**) demonstrated such selectivity (expressed as the selectivity index (SI), Table 6). The strongest selectivity against the MDA-MB-231 cell line was demonstrated by cladolosides H_1_ (**26**) (SI 8.1) and D (**18**) (SI 6.7) with hexasaccharide chains without sulfate groups and differing from each other by two monosaccharides in the upper semi-chain and by the presence/absence of a 25(26)-double bond (Appendix A).

### 2.3. Correlational Analysis and QSAR Model

The quantitative structure–activity relationship (QSAR) approach was applied to identify the statistically significant dependence between the hemolytic activity values and the structural peculiarities of the 26 glycosides isolated from *C. schmeltzii.* First, three-dimensional models were created for the glycosides, protonated at pH 7.4, and then energy was minimized. A search for dominant glycoside conformations was performed by LowModeMD Search method using the Molecular Operating Environment software (MOE, version 2020.09, Chemical Computing Group ULC, Montreal, QC, Canada) as well as some properties for 49 dominant glycoside conformers were estimated (SI, clad conformer properties.xlsx and coordinates for cladoloside conformers conf.xyz). For the analysis, the set of various 2D and 3D descriptors (total number—315) reflecting the physicochemical properties, energy values and topological indexes expressing the geometric properties of molecular structures, was calculated and studied using the QuaSAR-Descriptor tool of the MOE 2020.0901 CCG software (version 2020.09, Chemical Computing Group ULC, Montreal, QC, Canada).

It is important to emphasize that selecting specific descriptors is crucial because no single, universal set of descriptors has been found that can adequately describe all types of activities and properties of the compounds being analyzed. Therefore, suitable descriptors must be carefully selected for each type of activity and compound being studied. So, in addition to the descriptors characterizing the physicochemical properties of the glycosides (polarizability, refractive index, surface charge distribution, dipole moment, hydrogen bonds’ potential strength (donors and acceptors), hydrophobic volume, surface area, atomic valence connectivity index, etc.), following descriptors as the carbohydrate chain length and branching, type of the third and fifth sugar residue (xylose, glucose or quinovose), the sulfate groups’ positions were added to the descriptors set provided by the MOE-QuaSAR-Descriptor software (MOE, version 2020.09, Chemical Computing Group ULC, Montreal, QC, Canada). The correlational analysis revealed a direct positive correlation between the hemolytic activities of the tested compounds in vitro and such descriptors as the molecular refractivity, number of rings, atomic connectivity index [31], and partial charges distribution on the van der Waals surface, principal moment of inertia describing the different aspects of molecular shape. In contrast, the hydrophilic integy moment, lowest hydrophobic energy, critical packing parameter, and approximation of the sum of the VDW surface areas of hydrophobic atoms (Å^2^) were found to correlate negatively.

Principal component analysis (PCA) reduced the number of descriptors to those that made a significant contribution. As a result, glycosides were divided into two groups (Figure 2), indicating that the choice of descriptors was correct. Next, a PLS linear QSAR model was created using the QuaSAR-Model tool of the MOE (version 2020.09, Chemical Computing Group ULC, Montreal, QC, Canada) using these descriptors. The resulting model demonstrated excellent agreement with the experimental results for glycoside hemolytic activities, with a correlation coefficient r^2^ = 0.9854 and RMSE = 0.0813 (Appendix A). The quality check of the model confirmed its reliability, with a cross-validation correlation coefficient of r^2^_cros_ = 0.7286 and RMSE_cros_ = 0.2676. The QSAR model includes 107 terms, of which 70 have the maximum weight; however, a reduction in the number of descriptors below a given critical level leads to a decrease in the accuracy of the created model. All these data once again highlighted the complexity of the relationships between the structural characteristics of glycosides and their membranolytic action, due to the summative effect of the many different contributions of individual descriptors that have a significant influence in a multifactor model.

Results of the molecular modeling demonstrate that the membranolytic action of the glycosides containing 18(20)-lactone, 9(11)-double bond, and normal non-shortened side chain in the aglycones (features critically important for high hemolytic activity) also depends on the presence/absence/position of the side chain double bonds, the presence/absence and type of substituent at C-22 (OAc or OH groups), and the type of substituent at C-16 (OAc, OH, or oxo-groups). In addition, the length, branching, and monosaccharide composition of the carbohydrate chain, as well as the localization of sulfate groups, have a significant effect on activity.

The constructed QSAR model confirmed the fundamental importance of the presence of the 25(26)-double bond in the aglycone side chain and the O-acetic or oxo-group at C-22 and C-16, respectively, for the hemolytic activity of the studied glycosides. In contrast, the presence of a hydroxyl group at C-16, C-22, or C-25, as well as 23(24)- or 24(25)-double bonds, contributes negatively to the hemolytic activity of glycosides. Additionally, the total number of oxygen atoms, as well as donors and acceptors of hydrogen bonds, significantly contributes to the activity of glycosides, whereas the presence of OH-groups in the aglycones has a strong negative effect.

The observed structure–activity relationships based on hemolytic activity demonstrated that the most active compound was cladoloside M_1_ (**35**) from the tested series. In addition to the above-mentioned structural features (such as 18(20)-lactone and 25(26)-double bond), it has a keto-group at C-16 and a side chain without any functional groups at C-22 in combination with a hexasaccharide chain with glucose as the third and fifth residues. As shown by correlation analysis, these elements were strongly positively correlated with hemolytic activity.

The large number of parameters in the constructed QSAR model reflects the varying influence of structural elements on activity, depending on their location within the molecule. Thus, the distribution of polar and hydrophobic regions on the surface of the molecule, as well as the direction and magnitude of the dipole moment and the ratio of the components of the moment of inertia, are positively correlated with the hemolytic activity of glycosides. This is consistent with the high hemolytic activity of compounds containing bulky penta- and hexasaccharide chains and is illustrated by the high activity of pentaoside cladoloside B_2_ (**12**) and hexaosides, cladolosides C (**13**), D (**18**), and G (**25**) in comparison with the moderate activity of cladoloside A_2_ (**6**), which has the same aglycone but a tetrasaccharide chain. However, not only the architecture of carbohydrate chains but also the combination of monosaccharides and variability of their sequence lead to slightly different orientations of monosaccharide residues relative to each other, resulting in differences in their molecular packing density, which ultimately affects the activity.

The dual nature of the correlations of molecular volume and shape with hemolytic activity probably indicates the importance of the shape of the molecule as a whole, revealing the role of its individual parts. For example, the combination of a bulky hexasaccharide chain with two glucose residues (in the third and fifth positions) and a less bulky aglycone, due to the absence of *O*Ac-groups in the highly active cladoloside M_1_ (**35**), makes the final volume and shape of the molecule optimal for membranotropic action. Additionally, the low hemolytic activity of cladoloside A_3_ (**7**) is probably due to the non-optimal molecular volume and shape due to the presence of a tetrasaccharide chain with xylose as a third residue and the aglycone without any substituents in the side chain, in contrast to the more active and bulkier tetraoside cladoloside A_6_ (**10**), which has a third glucose in the sugar chain and a 22*-O*Ac- group. This observation is also consistent with the QSAR model, indicating the importance of charge distribution on the surface of the molecule and the significance of the density of packing of the carbohydrate moiety for the membranotropic action of glycosides.

Previously, it was shown that sulfate groups have an ambiguous effect on glycoside activity [16,20]. In this case, it was possible to compare the activities of monosulfated and non-sulfated penta- and hexaosides. The QSAR model demonstrated a significant contribution of the sulfate group at the fourth residue of the carbohydrate chain. The presence of a sulfate group at C-3 of this residue was shown to have a positive effect on the activity, as demonstrated by cladolosides T (**3**) and T_1_ (**4**). In general, the sulfate groups at the fourth sugar did not negatively impact the hemolytic activity of penta- or hexaosides.

SAR and QSAR analyses revealed significant differences in the effects of triterpene glycosides on different cell lines. Interestingly, the hemolytic activity of the studied glycosides was slightly correlated with cytotoxicity against healthy cells (MCF-10A) and significantly correlated with cytotoxicity against TNBC cells of the MDA-MB-231 line, which is reflected in the selectivity index (Table 6).

An extensive analysis of the main descriptors’ (43 variables) significance and influence in the correlational matrix (Appendix A) revealed that a number of structural features have an opposite influence, mainly in relation to the estrogen-dependent MCF-7 cell line and the triple-negative MDA-MB-231 cells. This is probably due to different membrane interactions of glycosides. The key drivers of activity in different cell lines were shown to be the *O*Ac-groups in the aglycones, sulfate groups, and glucose residues in the third and fifth positions in the sugar chains. The most pronounced antagonistic effects against MCF-7 and MDA-MB-231 cells were caused by the following descriptors: O-acetate or keto-group at C-16 of the aglycones, glucose residues in the fifth and sixth positions, the presence of 3*-O*-methyl groups at the fourth and sixth monosaccharide residues in carbohydrate chains, the presence of sulfate groups, and the localization of the sulfate group at C-3 of the fourth sugar unit.

These results indicate that numerous structural features of glycosides have different effects on cytotoxicity against hormone-dependent breast cancer cells MCF-7 and triple-negative MDA-MB-231 cells, suggesting that the glycosides target different membrane components in these tumor cell lines.

## 3. Materials and Methods

### 3.1. General Experimental Procedures

PerkinElmer 343 Polarimeter (PerkinElmer, Waltham, MA, USA) was used for specific rotation measuring; NMR spectra were registered on Avance III 700 Bruker FT-NMR spectrometer (Bruker BioSpin GmbH, Rheinstetten, Germany) (700.13/176.04 MHz (^1^H/^13^C, 30 °C, *δ*_C_ 148.9 resonance of C_5_D_5_N for ^13^C and *δ*_H_ 7.21 resonance of C_5_D_5_N for ^1^H used as the references, BBO probe)); ESI MS (negative ion mode) spectra were obtained on Agilent 6510 Q-TOF apparatus (Agilent Technology, Santa Clara, CA, USA), sample concentration 0.01 mg/mL; HPLC was conducted on Agilent 1260 Infinity II equipped with a differential refractometer (Agilent Technology, Santa Clara, CA, USA); columns were used: Supelco Discovery HS F5-5 (10 × 250 mm, 5 μM) (flow rate of 1.5 mL/min) and Supelcosil LC-Si (4.6 × 150 mm, 5 μM) (flow rate of 1.0 mL/min) (Supelco, Bellefonte, PA, USA).

### 3.2. Animals and Cells

Specimens of the sea cucumber *Cladolabes schmeltzii* (family Sclerodactylidae; order Dendrochirotida) were collected in the Nha Trang Gulf of the East Sea of Vietnam. Sampling was performed in May 2012 (collector I.Yu. Dolmatov) at a depth of 4−6 m. Sea cucumbers were identified taxonomically by I.Yu. Dolmatov; voucher specimens are preserved in the collection of the Museum of A.V. Zhirmunsky National Scientific Center of Marine Biology, Vladivostok, Russia.

Normal and cancer human cells were purchased from the Station of Blood Transfusion, Vladivostok, Russia (erythrocytes), ATCC (Manassas, VA, USA) (MCF-10A CRL-10317, T-47D HTB-133, MCF-7 HTB-22, and MDA-MB-231 CRM-HTB-26). Culturing conditions: RPMI-1640 medium with 1% penicillin/streptomycin and 10% fetal bovine serum (FBS) (Biolot, St. Petersburg, Russia) for T-47D cell line; Minimum Essential Medium (MEM) with 1% penicillin/streptomycin sulfate, and FBS (Biolot, St. Petersburg, Russia) to a final concentration of 10% for MCF-7 and MDA-MB-231 cells; DMEM/F12 medium with 20% FBS, 20 ng/mL EGF, 0.5 mg/mL hydrocortisone, 100 ng/mL cholera toxin, 10 μg/mL insulin, and 1% penicillin/streptomycin (Bioinnlabs, Rostov-on-Don, Russia) for MCF-10A cell line.

### 3.3. Extraction and Isolation

Sea cucumbers were minced and extracted twice with refluxing 60% EtOH. The dry weight of the residue is approximately 12.4 g. The combined extracts were concentrated to dryness in vacuo, dissolved in H_2_O, and chromatographed on a Polychrom-1 column (powdered Teflon, Biolar, Latvia) to eliminate inorganic salts and impurities. After this step 1.06 g of crude glycoside fraction was obtained, which was subsequently submitted to silica gel column chromatography using stepped gradient of solvent systems with different ratios of CHCl_3_/EtOH/H_2_O (from 25:12.5:1 to 2:3:1). This resulted in isolation of several glycosidic subfractions, including minor ones, weighing in 20 and 14 mg, and isolated with the most polar system (CHCl_3_/EtOH/H_2_O, 2:3:1). They were further separated by HPLC.

High-pressure liquid chromatography (HPLC) of the first subfraction (20 mg) on a reversed-phase column Supelco Discovery HS F5-5 (10 × 250 mm) with CH_3_CN/H_2_O/NH_4_OAc (1M water solution) in a ratio of (45/54/1) as the mobile phase yielded two fractions (1 and 2). Each of them was re-chromatographed on the same column but with ratio of the solvents (38/60/2) resulted in the isolation of cladoloside S_1_ (**2**) (3.5 mg) from fraction 1; and cladoloside S (**1**) (3 mg) from fraction 2. The second subfraction (14 mg) was subjected to HPLC on the same column with CH_3_CN/H_2_O/NH_4_OAc (1M water solution) (42/56/2) as the mobile phase and gave cladoloside T (**3**) (8 mg) and another fraction, which was subsequently subjected to HPLC on silica-based column Supelcosil LC-Si (4.6 × 150 mm) column with CHCl_3_/MeOH/H_2_O (85/20/1) as mobile phase to give individual cladoloside T_1_ (**4**) (3.2 mg).

#### 3.3.1. Cladoloside S (**1**)

Colorless powder; [α]_D_^20^−25° (*c* 0.1, H_2_O), mp 195 °C. Data of NMR: Table 1 and Table 2, Appendix A. (−)HR-ESI-MS *m*/*z*: 1367.5580 (calc. 1367.5584) [M_Na_ − Na]^−^, 683.2766 (calc. 683.2755) [M_Na_ − Na − H]^2−^; (−)ESI-MS/MS *m*/*z*: 1307.5 [M_Na_ − Na − CH_3_COOH]^−^, 1247.5 [M_Na_ − Na − 2CH_3_COOH]^−^, 1235.5 [M_Na_ − Na − Xyl (C_5_H_8_O_4_)]^−^, 1175.5 [M_Na_ − Na − Xyl (C_5_H_8_O_4_) − CH_3_COOH]^−^, 797.2 [M_Na_ − Na − Agl (C_34_H_49_O_7_) − H]^−^, 533.1 [M_Na_ − Na − Agl (C_34_H_49_O_7_) − 2Xyl (2C_5_H_8_O_4_)]^−^, and 387.0 [M_Na_ − Na − Agl (C_34_H_49_O_7_) − 2Xyl (2C_5_H_8_O_4_) − Qui (C_6_H_10_O_4_)]^−^.

#### 3.3.2. Cladoloside S_1_ (**2**)

Colorless powder; [α]_D_^20^−20° (*c* 0.1, H_2_O), mp 202 °C. Data of NMR: Table 3 and Appendix A. (−)HR-ESI-MS *m*/*z*: 1369.5735 (calc. 1369.5740) [M_Na_ − Na]^−^, 684.2849 (calc. 684.2834) [M_Na_ − Na − H]^2−^; (−)ESI-MS/MS *m*/*z*: 1309.5 [M_Na_ − Na − CH_3_COOH]^−^, 1249.5 [M_Na_ − Na − 2CH_3_COOH]^−^, 1177.5 [M_Na_ − Na − Xyl − CH_3_COOH]^−^, 797.2 [M_Na_ − Na − Agl (C_34_H_51_O_7_) − H]^−^, 533.1 [M_Na_ − Na − Agl (C_34_H_51_O_7_) − 2Xyl (2C_5_H_8_O_4_)]^−^, and 387.0 [M_Na_ − Na − Agl (C_34_H_51_O_7_) − 2Xyl (2C_5_H_8_O_4_) − Qui (C_6_H_10_O_4_)]^−^.

#### 3.3.3. Cladoloside T (**3**)

Colorless powder; [α]_D_^20^−28° (*c* 0.1, H_2_O), mp 215 °C. Data of NMR: Table 4 and Appendix A. (−)HR-ESI-MS *m*/*z*: 1559.6233 (calc. 1559.6217) [M_Na_ − Na]^−^; 779.3096 (calc. 779.3072) [M_Na_− Na − H]^2−^; (−)ESI-MS/MS *m*/*z*: 1499.6 [M_Na_ − Na − CH_3_COOH]^−^, 1439.6 [M_Na_ − Na − 2CH_3_COOH]^−^, 1383.6 [M_Na_ − Na − MeGlc (C_7_H_12_O_5_)]^−^, 1323.5 [M_Na_ − Na − MeGlc (C_7_H_12_O_5_) − CH_3_COOH]^−^, 1161.5 [M_Na_ − Na − CH_3_COOH − MeGlc (C_7_H_12_O_5_) − Glc (C_6_H_10_O_5_)]^−^, 1101.5 [M_Na_ − Na − 2CH_3_COOH − MeGlc (C_7_H_12_O_5_) − Glc (C_6_H_10_O_5_)]^−^, 1007.3 [M_Na_ − Na − Agl (C_34_H_49_O_6_) + H]^−^, 519.1 [M_Na_ − Na − Agl (C_34_H_49_O_6_) − MeGlc (C_7_H_13_O_6_) − Glc (C_6_H_10_O_5_) − Xyl (C_5_H_8_O_4_)]^−^.

#### 3.3.4. Cladoloside T_1_ (**4**)

Colorless powder; [α]_D_^20^−24° (*c* 0.1, H_2_O), mp 206 °C. Data of NMR: Appendix A. (−)HR-ESI-MS *m*/*z*: (−)HR-ESI-MS *m*/*z*: 1561.6377 (calc. 1561.6377) [M_Na_ − Na]^−^; (−)ESI-MS/MS *m*/*z*: 1501.6 [M_Na_ − Na − CH_3_COOH]^−^, 1367.6 [M_Na_ − Na − MeGlc (C_7_H_13_O_6_) − H]^−^, 1325.5 [M_Na_ − Na − MeGlc (C_7_H_12_O_5_) − CH_3_COOH]^−^, 1163.5 [M_Na_ − Na − CH_3_COOH − MeGlc (C_7_H_12_O_5_) − Glc (C_6_H_10_O_5_)]^−^, 1103.5 [M_Na_ − Na − 2CH_3_COOH − MeGlc (C_7_H_12_O_5_) − Glc (C_6_H_10_O_5_)]^−^, 1007.3 [M_Na_ − Na − Agl (C_34_H_51_O_6_) − H]^−^, 519.1 [M_Na_ − Na − Agl (C_34_H_49_O_6_) − MeGlc (C_7_H_13_O_6_) − Glc (C_6_H_10_O_5_) − Xyl (C_5_H_8_O_4_)]^−^.

### 3.4. Cytotoxic Activity (MTT Assay)

Solutions (20 µL) of the tested substances at different concentrations (0.1–20 µM) or positive controls—cisplatin and cucumarioside A_0_-1 [15] and the cell suspension (180 µL) (MCF-10A, MCF-7, T-47D, and MDA-MB-231—7 × 10^3^ cells per well) were added to the wells of 96-well plates and incubated overnight at 37 °C and 5% CO_2_, then the cells were precipitated by centrifugation, the medium was stowed and 100 µL of fresh medium and 10 µL of 3-(4,5-dimethylthiazol-2-yl)-2,5-diphenyltetrazolium bromide (MTT) (PanReac, AppliChem, Darmstadt, Germany) stock solution (5 mg/mL) were added into each well. After 4 h of incubation and addition of 100 µL of SDS-HCl, cells were subsequently incubated at 37 °C for 18 h. A Multiskan FC microplate photometer (Thermo Fisher Scientific, Waltham, MA, USA) was used to measure the absorbance of the converted dye formazan at 570 nm. The cytotoxic activity of each glycoside was expressed as the concentration that caused 50% inhibition of cell metabolic activity (IC50). The experiments were conducted in triplicate, *p* < 0.05.

### 3.5. Hemolytic Activity

A total of 450 g of human blood (B(III) Rh+) was used to obtain erythrocytes by centrifuging three times for 5 min with phosphate-buffered saline (PBS) (pH 7.4) at 4 °C on a centrifuge LABOFUGE 400R (Heraeus, Hanau, Germany). Ice-cold PBS (pH 7.4) was used for resuspension of erythrocyte residue to a final optical density of 1.5 at 700 nm. Then, 20 µL of tested compound solutions or control (cucumarioside A_0_-1 [15]) were added to 180 µL of erythrocyte suspension in V-bottom 96-well plates and incubated for 1 h at 37 °C. After centrifugation (900 g, 10 min, laboratory centrifuge LMC-3000 (Biosan, Riga, Latvia)), 100 µL of supernatant was transferred into new flat plates. The values of erythrocyte lysis were measured on microplate photometer Multiskan FC (Thermo Fisher Scientific, Waltham, MA, USA) at λ = 570 nm as hemoglobin concentration in supernatant. The effective dose, causing lysis of 50% erythrocytes (ED50), was calculated with SigmaPlot software (version 14, Grafiti LLC, Palo Alto, CA, USA). All the experiments were carried out in triple repetitions, *p* < 0.05.

### 3.6. Building a QSAR Model

QSAR model for the set of 26 glycosides was built using the QuaSAR-Descriptor and QuaSAR-Model tools of MOE (version 2020.09, Chemical Computing Group ULC, Montreal, QC, Canada). The order of stages of Preliminary Data Processing was as follows: the charge distribution calculation, optimization of the glycoside geometric structure, search for preferred spatial configurations of glycosides by LowModeMD Search method [32], which generated conformations using a short ~1 ps run of Molecular Dynamics (MD) with velocities initialized to low-frequency vibrational modes at constant temperature [33], followed by an all-atom energy minimization and determination of numerical characteristics reflecting the physicochemical properties of these glycoside molecules—2D and 3D descriptors calculation. The next stage of the Data Analysis involves the study of statistical relationships between the calculated descriptors and biological indicators of glycoside activity expressed as pED_50_, the PCA method application to reduce the number of independent variables, while maintaining the maximum dispersion of activity, elimination of multicollinear descriptors, building a QSAR model and the assessment of model’s reliability (cross-validation). Then, the Model Optimization and Final Testing were performed by removing the descriptors not contributing to the model, and model checking by making a graph showing the correlation between the model-predicted value and the experimental activity value expressed as pED_50_.

## 4. Conclusions

As a result of the investigation of the glycosidic composition of the sea cucumber *Cladolabes schmeltzii*, 44 glycosides, cladolosides of different groups, consisting of 21 types of carbohydrate chains and 9 diverse aglycones, and constituting a combinatorial library, were isolated and characterized.

Structurally biogenetic analysis of the carbohydrate chains comprising the cladolosides of groups A–T (**1**–**44**) showed that sulfation in most cases is a final stage of biosynthesis. Based on the structures of the recently discovered cladolosides T (**3**) and T_1_ (**4**), it can be assumed that the enzymatic processes of sulfation and O-methylation compete with each other. This may be due to the high level of expression of the enzymes and lack of compartmentalization inside the cells of organisms–producers that leads to a certain degree of randomness in enzymatic reactions, while providing an advantage in terms of chemical diversity of the glycosides.

Analysis of the membranolytic activity of cladolosides of different groups showed that hexaosides, which are the final products of biosynthesis, exhibited higher cytotoxic activity than their precursors, pentaosides. Cladolosides of the group A (tetraosides) are weak cytotoxins and, therefore, are not synthesized in large quantities.

Analysis of structure–activity relationships based on the hemolytic activity of glycosides showed that the sulfate group attached instead of the 3*-O-*Me group to the terminal sugar residue does not decrease activity. Two glycosides from *C. schmeltzii* display selectivity of action toward triple-negative breast cancer cells MDA-MB-231, being non-toxic in relation to normal mammary cells MCF-10A.

QSAR analysis of *C. schmeltzii* glycosides confirmed the complex nature of the revealed patterns due to the combined influence of a huge number of descriptors on the membranolytic activity of glycosides. Moreover, numerous structural features of glycosides are related in different ways to cytotoxicity against hormone-dependent breast cancer cells MCF-7 and triple-negative MDA-MB-231 cells, suggesting that the glycosides target different membrane components in these tumor cell lines. Therefore, the application of QSAR for the prediction or modeling of the biological activity of triterpene glycosides in sea cucumbers is prospective and useful.

## Figures and Tables

**Figure 1 marinedrugs-23-00265-f001:**
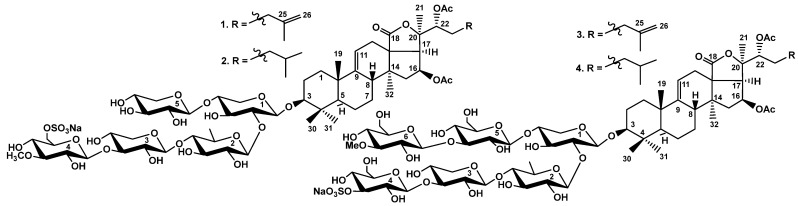
Chemical structures of the glycosides from *Cladolabes schmeltzii*: **1**—cladoloside S; **2**—cladoloside S_1_; **3**—cladoloside T; **4**—cladoloside T_1_.

**Figure 2 marinedrugs-23-00265-f002:**
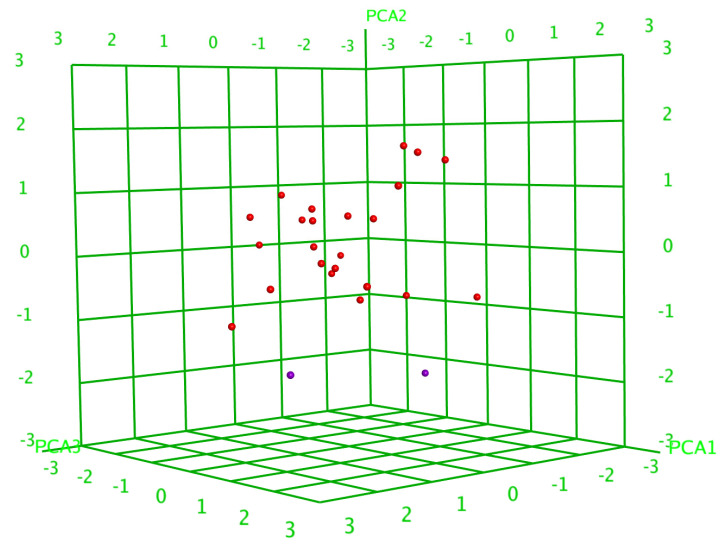
Three-dimensional plot of hemolytic activity (pED_50_) depending on the principal component values (PCA1—PCA3) calculated for 26 glycosides. The glycosides that demonstrated hemolytic activity with ED_50_ ≤ 10 µM are outlined as active and are marked in red, while the rest are marked in violet.

**Table 1 marinedrugs-23-00265-t001:** One- and two-dimensional NMR data of the aglycone moiety of cladoloside S (**1**).

Position	*δ*_C_, mult. ^a^	*δ*_H_, mult. (*J* in Hz) ^b^	HMBC	ROESY
1	36.1, CH_2_	1.81, m		H-11
		1.44, m		H-3, H-5, H-11
2	26.9, CH_2_	2.22, m		
		1.97, m		H-19, H-30
3	88.6, CH	3.25, dd (12.0; 3.9)	C: 4, 30, 31, C:1 Xyl1	H-1, H-5, H-31, H1-Xyl1
4	39.8, C			
5	52.7, CH	0.92, m	C: 6, 19, 30, 31	H-1, H-3, H-31
6	20.7, CH_2_	1.71, m		
		1.53, m		H-8, H-19, H-30
7	27.8, CH_2_	1.64, m		H-15, H-32
		1.24, m		
8	39.5, CH	3.25, brd (13.0)		H-6, H-15, H-19
9	150.9, C			
10	39.3, C			
11	110.4, CH	5.23, m	C: 10, 13	
12	33.9, CH_2_	2.53, brdd (17.2; 6.1)	C: 9, 11, 14, 18	H-17, H-32
		2.50, brd (17.2)	C: 9, 11, 14, 18	H-21
13	59.0, C			
14	43.6, C			
15	44.4, CH_2_	2.40, dd (12.1; 6.1)	C: 13, 14, 16, 17, 32	H-7, H-32
		1.40, brdd (12.1; 9.1)	C: 14, 16, 32	H-8
16	75.2, CH	5.90, brq (9.1)	C: 13, 20, OAc-16	
17	53.6, CH	2.91, d (9.1)	C: 12, 13, 15, 16, 18, 20, 21	H-12, H-21, H-32
18	176.0, C			
19	22.0, CH_3_	1.35, s	C: 1, 5, 9, 10	H-1, H-2, H-6, H-8, H-30
20	85.8, C			
21	22.4, CH_3_	1.64, s	C: 17, 20, 22	H-12, H-17
22	74.7, CH	6.66, d (11.1)	C: 20, 21, OAc-22	
23	29.3, CH_2_	1.78, m		
24	34.3, CH_2_	2.26, m	C: 25	
		2.16, m		H-22
25	144.9, C			
26	110.6, CH_2_	4.84, brs	C: 24, 25, 27	
		4.83, brs	C: 24, 25, 27	
27	22.2, CH_3_	1.74, s	C: 24, 25, 26	H-26
30	16.5, CH_3_	1.11, s	C: 3, 4, 5, 31	H-2, H-6, H-19, H-31
31	27.9, CH_3_	1.29, s	C: 3, 4, 5, 30	H-3, H-5, H-6, H-30, H-1Xyl1
32	21.0, CH_3_	0.91, s	C: 8, 13, 14, 15	H-7, H-12, H-15, H-16, H-17
OAc-16	169.4, C			
	21.4, CH_3_	2.28, s	C: 16	H-22
OAc-22	170.3, C			
	20.7, CH_3_	2.08, s	C: 22	

^a^ Recorded at 176.04 MHz in C_5_D_5_N/D_2_O (4/1). ^b^ Recorded at 700.13 MHz in C_5_D_5_N/D_2_O (4/1). The original spectra of **1** are provided in Appendix A.

**Table 2 marinedrugs-23-00265-t002:** One- and two-dimensional NMR data of the carbohydrate chain of cladoloside S (**1**).

Atom	*δ*_C_, mult. ^a,b,c^	*δ*_H_, mult. (*J* in Hz) ^d^	HMBC	ROESY
Xyl1 (1→C-3)				
1	105.1, CH	4.76, d (7.5)	C-3	H-3; H-3, 5 Xyl1
2	**83.4**, CH	4.09, t (8.8)	C: 1 Xyl1, 1 Qui2	H-1 Qui2
3	75.3, CH	4.22, t (8.8)	C: 2 Xyl 1	
4	**76.2**, CH	4.29, m	C: 1 Xyl5	
5	64.0, CH_2_	4.40, dd (11.9; 4.4)	C: 1 Xyl1	
		3.63, brdd (11.9; 9.4)		H-1, 3 Xyl1
Qui2 (1→2Xyl1)				
1	105.4, CH	5.17, d (7.5)	C: 2 Xyl1	H-2 Xyl1, H-3, 5 Qui2
2	76.4, CH	4.05, t (9.2)	C: 1, 3 Qui2	
3	75.1, CH	4.12, t (9.2)	C: 2 Qui2	H-1 Qui2
4	**85.6**, CH	3.67, t (9.2)	C: 3, 5 Qui2, 1 Xyl3	H-1 Xyl3, H-2 Qui2
5	71.6, CH	3.82, dd (9.2; 5.9)		H-1 Qui2
6	18.1, CH_3_	1.76, d (5.9)	C: 4, 5 Qui2	
Xyl3 (1→4Qui2)				
1	104.6, CH	4.87, d (7.9)	C: 4 Qui2	H-4 Qui2, H-3, 5 Xyl3
2	73.5, CH	3.97, t (8.9)	C: 1, 3 Xyl3	
3	**86.6**, CH	4.14, t (8.9)	C: 2, 4 Xyl3; 1 MeGlc4	H-1 MeGlc4, H-1, 5 Xyl3
4	68.7, CH	4.05, m		
5	66.3, CH_2_	4.20, dd (11.6; 4.8)	C: 3, 4 Xyl3	
		3.64, brt (11.6)	C: 1 Xyl3	H-1 Xyl3
MeGlc4 (1→3Xyl3)				
1	105.0, CH	5.28, d (8.5)	C: 3 Xyl3	H-3 Xyl3, H-3, 5 MeGlc4
2	74.4, CH	3.85, t (8.5)	C: 1 MeGlc4	
3	87.1, CH	3.64, t (8.5)	C: 2, 4 MeGlc4, OMe	
4	70.0, CH	4.19, t (8.5)		
5	76.3, CH	4.01, m		H-1 MeGlc4
6	*66.6*, CH_2_	5.01, m	C: 4, 5 MeGlc4	
OMe	60.5, CH_3_	3.81, s	C: 3 MeGlc4	H-3 MeGlc4
Xyl5 (1→4Xyl1)				
1	103.5, CH	4.91, d (8.0)	C: 4 Xyl1	H-4 Xyl1, H-3, 5 Xyl5
2	73.7, CH	4.00, t (8.6)	C: 1, 3 Xyl5	
3	77.8, CH	4.12, t (8.6)	C: 2, 4 Xyl5	H-1 Xyl5
4	70.7, CH	4.18, m		
5	67.2, CH_2_	4.33, dd (11.7; 4.9)	C: 1, 3, 4 Xyl5	
		3.68, t (11.7)	C: 1, 3, 4 Xyl5	

^a^ Recorded at 176.04 MHz in C_5_D_5_N/D_2_O. ^b^ Bold—interglycosidic positions. ^c^ Italics—sulfate position. ^d^ Recorded at 700.13 MHz in C_5_D_5_N/D_2_O. Multiplicity by 1D TOCSY. The original spectra of **1** are provided in Appendix A.

**Table 3 marinedrugs-23-00265-t003:** One- and two-dimensional NMR data of the aglycone moiety of cladoloside S_1_ (**2**).

Position	*δ*_C_, mult. ^a^	*δ*_H_, mult. (*J* in Hz) ^b^	HMBC	ROESY
1	36.1, CH_2_	1.70, m		H-19
		1.31, m		H-3, H-5
2	26.7, CH_2_	2.07, m		
		1.85, m		H-19, H-30
3	88.8, CH	3.14, brd (11.7)	C:1 Xyl1	H-1, H-5, H-31, H1-Xyl1
4	39.7, C			
5	52.7, CH	0.78, brd (11.7)		H-3, H-31
6	21.0, CH_2_	1.57, m		
		1.42, m		
7	27.7, CH_2_	1.60, m		
		1.15, m		H-5, H-32
8	39.5, CH	3.12, brd (16.0)		H-15
9	150.6, C			
10	39.2, C			
11	110.7, CH	5.18, m		H-1
12	33.9, CH_2_	2.62, brd (16.8)		
		2.52, dd (16.8; 6.0)	C: 17	H-21
13	59.2, C			
14	43.6, C			
15	44.4, CH_2_	2.35, dd (12.0; 7.2)	C: 13, 17, 32	H-32
		1.28, brd (12.0)		H-8
16	75.5, CH	5.83, brq (9.6)		H-32
17	53.4, CH	2.97, d (9.6)	C: 12, 13, 18, 21	H-12, H-21, H-32
18	176.9, C			
19	22.0, CH_3_	1.23, s	C: 1, 5, 9, 10	H-1, H-30
20	86.5, C			
21	22.5, CH_3_	1.65, s	C: 17, 20, 22	
22	75.4, CH	6.48, d (10.8)		
23	28.5, CH_2_	1.52, m		
24	35.1, CH_2_	1.29, m		
		1.15, m		
25	28.1, CH	1.52, m		
26	22.5, CH_3_	0.83, s	C: 24, 25, 27	
27	22.8, CH_3_	0.82, s	C: 24, 25, 26	H-31
30	16.5, CH_3_	0.98, s	C: 3, 4, 5, 31	H-2, H-6, H-19, H-30,
31	27.9, CH_3_	1.16, s	C: 3, 4, 5, 30	H-1 Xyl1
32	21.0, CH_3_	0.91, s	C: 8, 13, 14, 15	H-12, H-15, H-17, H-24
OAc-16	171.4, C			
	20.9, CH_3_	2.07, s		
OAc-22	170.5, C			
	21.6, CH_3_	2.25, s		H-22, H-24, H-26, H-27

^a^ Recorded at 176.04 MHz in C_5_D_5_N/D_2_O. ^b^ Recorded at 700.13 MHz in C_5_D_5_N/D_2_O. The original spectra of **2** are provided in Appendix A.

**Table 4 marinedrugs-23-00265-t004:** One- and two-dimensional NMR data of the carbohydrate moiety of cladoloside T (**3**).

Atom	*δ*_C_, mult. ^a,b,c^	*δ*_H_, mult. (*J* in Hz) ^d^	HMBC	ROESY
Xyl1 (1→C-3)				
1	104.8, CH	4.66, d (6.9)	C-3	H-3; H-3, 5 Xyl1
2	**82.5**, CH	3.96, t (7.6)	C: 1,3 Xyl1, C: 1 Qui2	H-1 Qui2; H-4 Xyl1
3	75.1, CH	4.16, t (7.6)	C: 2,4 Xyl 1	H-1 Xyl1
4	**77.1**, CH	4.22, m	C: 3 Xyl1, C: 1 Glc5	H-1 Glc5
5	63.5, CH_2_	4.39, m	C: 1, 3 Xyl1	
		3.63, m		H-1 Xyl1
Qui2 (1→2Xyl1)				
1	104.7, CH	5.02, d (7.7)	C: 2 Xyl1	H-2 Xyl1, H-3, 5 Qui2
2	75.7, CH	3.87, t (8.7)	C: 1, 3 Qui2	H-4 Qui2
3	74.8, CH	3.94, t (8.7)	C: 2, 4 Qui2	H-1 Qui2
4	**85.7**, CH	3.49, t (8.7)	C: 3, 5 Qui2, 1 Xyl3	H-1 Xyl3, H-2 Qui2
5	71.5, CH	3.67, dd (8.7; 6.8)		H-1 Qui2
6	17.8, CH_3_	1.63, d (6.8)	C: 4, 5 Qui2	
Xyl3 (1→4Qui2)				
1	104.4, CH	4.76, d (8.3)	C: 4 Qui2	H-4 Qui2, H-3, 5 Xyl3
2	73.6, CH	3.88, t (8.3)	C: 1, 3 Xyl3	
3	**86.5**, CH	4.12, t (8.3)	C: 2, 4 Xyl3; 1 Glc4	H-1 Glc4, H-1 Xyl3
4	68.6, CH	3.94, m		
5	65.9, CH_2_	4.09, dd (10.8; 5.0)	C: 1, 3, 4 Xyl3	
		3.60, t (10.8)	C: 1, 3, 4 Xyl3	H-1, 3 Xyl3
Glc4 (1→3Xyl3)				
1	104.0, CH	5.27, d (8.3)	C: 3 Xyl3	H-3 Xyl3, H-3, 5 Glc4
2	73.6, CH	3.98, t (8.3)	C: 1, 3 Glc4	
3	*84.2*, CH	5.03, t (8.3)	C: 2, 4 Glc4	H-1, 5 Glc4
4	70.0, CH	3.95, t (8.3)	C: 3, 5 Glc4	H-6 Glc4
5	77.1, CH	3.85, t (8.3)	C: 4 Glc4	H-1, 3 Glc4
6	61.7, CH_2_	4.33, dd (11.6; 2.1)	C: 4 Glc4	
		3.99, m	C: 5 Glc4	
Glc5 (1→4Xyl1)				
1	102.2, CH	4.92, d (6.8)	C: 4 Xyl1	H-4 Xyl1, H-3, 5 Glc5
2	73.3, CH	3.92, t (7.9)	C: 1 Glc5	
3	86.9, CH	4.19, t (7.9)	C: 2, 4 Glc5, C: 1 MeGlc6	H-1 MeGlc6; H-1, 5 Glc5
4	69.4, CH	3.87, t (7.9)	C: 3, 5 Glc5	
5	77.5, CH	3.87, t (7.9)	C: 6 Glc5	H-3 Glcl5
6	61.7, CH_2_	4.29, brd (11.7)		
		4.02, brd (11.7)	C: 5 Glc5	
MeGlc6 (1→3Glc5)				
1	104.5, CH	5.19, d (7.5)	C: 3 Glc5	H-3 Glc5; H-3, 5 MeGlc6
2	74.5, CH	3.85, t (9.1)	C: 1, 3 MeGlc6	
3	86.9, CH	3.67, t (9.1)	C: 2, 4 MeGlc6; OMe	H-1 MeGlc6
4	70.3, CH	3.87, t (9.1)	C: 3, 5 MeGlc6	
5	77.5, CH	3.93, t (9.1)	C: 6 MeGlc6	H-1 MeGlc6
6	61.7, CH_2_	4.37, brd (11.9)	C: 4 MeGlc6	H-4 MeGlc6
		4.04, dd (11.9; 6.0)	C: 5 MeGlc6	
OMe	60.7, CH_3_	3.80, s	C: 3 MeGlc6	

^a^ Recorded at 176.04 MHz in C_5_D_5_N/D_2_O. ^b^ Bold—interglycosidic positions. ^c^ Italics—sulfate position. ^d^ Recorded at 700.13 MHz in C_5_D_5_N/D_2_O. Multiplicity by 1D TOCSY. The original spectra of **3** are provided in Appendix A.

**Table 5 marinedrugs-23-00265-t005:** Cytotoxic activities of glycosides **1**–**4** and 26 known glycosides from *C. schmeltzii*, as well as cucumarioside A_0_-1 and cisplatin as positive controls, against human erythrocytes and MCF-10A, MCF-7, T-47D, and MDA-MB-231 human cell lines.

Glycosides (Cladolosides)	ED_50_, µM, Erythrocytes	Cytotoxicity, IC_50_ µM			
		MCF-10A	MCF-7	T-47D	MDA-MB-231
S (**1**)	3.40 ± 0.25	>20.00	>20.00	14.85 ± 0.71	13.24 ± 1.08
S_1_ (**2**)	4.78 ± 0.30	>20.00	>20.00	>20.00	>20.00
T (**3**)	0.39 ± 0.03	8.23 ± 0.52	8.99 ± 0.61	8.21 ± 0.72	3.41 ± 0.33
T_1_ (**4**)	0.12 ± 0.02	8.41 ± 0.40	7.78 ± 0.62	7.41 ± 0.69	2.94 ± 0.21
A_1_ (**5**)	6.17 ± 0.51	>20.00	>20.00	>20.00	>20.00
A_2_ (**6**)	2.78 ± 0.32	>20.00	>20.00	>20.00	>20.00
A_3_ (**7**)	7.58 ± 0.69	>20.00	>20.00	>20.00	>20.00
A_4_ (**8**)	13.60 ± 0.70	>20.00	>20.00	>20.00	>20.00
A_5_ (**9**)	7.19 ± 0.53	>20.00	>20.00	>20.00	>20.00
A_6_ (**10**)	3.43 ± 0.22	>20.00	>20.00	>20.00	>20.00
B_1_ (**11**)	0.55 ± 0.04	>20.00	>20.00	12.99 ± 1.07	10.12 ± 1.00
B_2_ (**12**)	0.31 ± 0.02	12.23 ± 1.20	14.52 ± 0.48	12.50 ±0.96	3.92 ± 0.35
C (**13**)	0.44 ± 0.05	>20.00	>20.00	12.39 ± 0.66	13.01 ± 0.80
C_1_ (**14**)	0.15 ± 0.02	>20.00	>20.00	>20.00	15.27 ± 0.71
C_3_ (**16**)	12.10 ± 0.58	>20.00	>20.00	>20.00	>20.00
D (**18**)	0.29 ± 0.03	>20.00	15.31 ± 0.94	14.93 ± 0.85	2.98 ± 0.24
G (**25**)	0.35 ± 0.04	>20.00	> 20.00	>20.00	3.95 ± 0.38
H_1_ (**26**)	0.21 ± 0.02	>20.00	15.93 ± 1.14	13.20 ± 0.72	2.47 ± 0.32
I_1_ (**28**)	1.72 ± 0.10	13.44 ± 0.92	11.23 ± 1.02	9.91 ± 0.68	3.02 ± 0.28
I_2_ (**29**)	4.24 ± 0.31	>20.00	>20.00	>20.00	>20.00
J_1_ (**30**)	1.10 ± 0.10	>20.00	>20.00	>20.00	6.18 ± 0.54
K_1_ (**31**)	0.31 ± 0.02	>20.00	>20.00	12.13 ± 1.00	5.83 ± 0.41
K_2_ (**32**)	4.63 ± 0.41	14.45 ± 1.07	>20.00	14.08 ± 0.91	6.81 ± 0.73
L_1_ (**33**)	0.20 ± 0.02	17.88 ± 1.12	12.70 ± 1.01	9.03 ± 0.81	13.36 ± 1.08
M_1_ (**35**)	0.08 ± 0.01	9.81 ± 0.86	>20.00	17.14 ± 1.52	5.05 ± 0.38
P_2_ (**41**)	0.17 ± 0.02	11.10 ± 0.74	16.32 ± 1.03	10.59 ± 0.61	6.92 ± 0.44
Cuc A_0_-1	1.63 ± 0.21	7.51 ± 0.63	11.33 ± 1.00	8.46 ± 0.53	3.53 ± 0.54
cysplatin	-	82.33 ± 5.21	152.00 ± 8.02	≥160.00	>160.00

**Table 6 marinedrugs-23-00265-t006:** Selectivity index of the glycosides to different cancer cells (SI; a ratio of IC_50_ calculated for healthy (MCF-10A) and cancer cells).

Glycoside	Selectivity Index (SI)		
	MCF-7	T-47D	MDA-MB-231
Cladoloside S (**1**)	-	>1.35	>1.51
Cladoloside T (**3**)	0.92	1.00	2.41
Cladoloside T_1_ (**4**)	1.08	1.13	2.86
Cladoloside B_1_ (**11**)	-	>1.54	>1.98
Cladoloside B_2_ (**12**)	0.84	0.98	3.12
Cladoloside C (**13**)	-	>1.61	>1.54
Cladoloside C_1_ (**14**)	-	-	>1.13
Cladoloside D (**18**)	>1.31	>1.34	>**6.7**
Cladoloside G (**25**)	-	-	>5.06
Cladoloside H_1_ (**26**)	1.26	1.51	**8.10**
Cladoloside I_1_ (**28**)	1.20	1.36	4.45
Cladoloside J_1_ (**30**)	-	-	>3.24
Cladoloside K_1_ (**31**)	-	>1.65	>3.83
Cladoloside K_2_ (**32**)	-	1.03	2.17
Cladoloside L_1_ (**33**)	1.41	1.98	1.34
Cladoloside M_1_ (**35**)	-	0.57	1.94
Cladoloside P_2_ (**41**)	0.68	1.05	1.60
Cucumarioside A_0_-1	0.66	0.89	2.13

## Data Availability

The original contributions presented in this study are included in the article. Further inquiries can be directed to the corresponding author.

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
