# Peer review of "Cladolosides of Groups S and T: Triterpene Glycosides from the Sea Cucumber Cladolabes schmeltzii with Unique Sulfation; Human Breast Cancer Cytotoxicity and QSAR"

_marinedrugs, 2025, doi:10.3390/md23070265_

Round 1

Reviewer 1 Report

Comments and Suggestions for Authors

Silchenko and co-workers report four new monosulfated triterpene penta- and hexaglycosides from the Vietnamese sea cucumber Cladolabes schmelii, along with their cytotoxicities against a human breast cancer cell line. While the subject matter will be of interest to the marine natural products community and the journal’s readership, the manuscript requires major revisions before it can be reconsidered for publication. The standard of English throughout the manuscript is insufficient for clear scientific communication. It is strongly recommended that the manuscript be thoroughly edited by a fluent English speaker or a professional scientific editor prior to resubmission. Several sections of the text have been highlighted in a marked pdf (see attach) and commented on below, but the grammatical issues are widespread and require a comprehensive review. Additionally, the authors have not numbered the newly reported compounds, which significantly hinders the reader’s ability to follow the structural discussion and assess the validity of the structure elucidation. Numbering each compound consistently in the text, figures, and tables is essential for clarity and proper scientific communication. I have included some comments below, but there are far more that will need to be addressed prior to resubmission to the journal. I recommend rejection of the manuscript until it undergoes major revision suitable for Marine Drugs.

Comments on the Quality of English Language

The English language used throughout the manuscript needs significant improvement.

Author Response

We are very grateful to the Reviewer for such in-depth analysis of our manuscript. We fully agree with the Reviewer's comments regarding formulas and atom numbering. Therefore, the formulas of the new glycosides have been corrected so that the structures are clearly represented, carbon atoms in aglycones and monosaccharide residues in carbohydrate chains are numbered. The English has been thoroughly edited throughout the manuscript; all the corrections are highlighted with yellow.

The responses to the Reviewer's comments are expressed below:

*Page 1, Title: the title of the manuscript is needlessly wordy and incoherent and as such should be improved and refined

The title is improved but we prefer to conserve the group names of new substances in order to avoid of a similarity of titles of different articles concerning the same sea cucumber species. The absence of such information in the title may confused the readers including the authors of future reviews.

*Page 2, Lines 52–55: Sentence needs grammar clarified and corrected

Fixed.

*Page 2, Line 58: correct ‘activity’ to ‘activities’

Fixed.

*Page 2, Lines 62–64: Sentence needs grammar clarified and corrected

Fixed.

*Page 2, Line 64: remove superfluous ‘the’

Fixed.

*Page 2, Line 65: remove superfluous ‘very’

Fixed.

*Page 2, Lines 68–80: Sentence needs grammar clarified and corrected

Fixed.

*Page 2, Line 83: change ‘synergetic’ to more commonly used ‘synergistic’

Fixed.

*Page 2, Lines 85–88: Sentence needs grammar clarified and corrected

Fixed.

*Page 2, Line 88: correct ‘As result’ to ‘As a result,’

This phrase (line 88) was deleted at text correction, but the error is corrected throughout the text.

*Page 3, Line 97: Sentence needs grammar clarified and corrected

Fixed.

*Page 3, Figure 1: atom numbers should be added to chemical structures consistent with that presented in NMR tables, the chemical structures need to be clarified for 1 and 2 compared with 3 and 4 with respect to R groups. For example, 1 and 2 are one carbohydrate residue shorter than 3 and 4 but it is not clear whether they are -OH or -OSO3NA etc. The R groups should be clearly labelled and easy for the reader to decipher.

Figure 1 is redrawn according to the Reviewer’s comment, position numbers of carbons are added, the meaning of R groups are clarified.

*Page 3, Table 1 heading: change ‘13C and 1H NMR chemical shifts and HMBC and ROESY correlations’ to ‘ One- and two-dimensional NMR data’ repeat for Tables 2 and 3 also.

Fixed.

*Page 15, Line 501: correct ‘1061 mg’ to ‘1.06 g’

Fixed.

*Page 15, Lines 503 and 505: simplify these ratios ie 2:3:1

Ratios are simplified.

*Page 15, Line 506: remove ‘of’

In the text “HPLC” is “High-pressure liquid chromatography”, so the preposition 'of' is necessary after word ‘chromatography’.

*Page 15, Lines 507-509: refer to isocratic mobile phase and ratios should read in % ie 45% CH3CN/54% H2O/1% NH4OAc

*Page 16, Line 515: refer to isocratic mobile phase and ratios should read in % ie 85% CH3CN/20% MeOH/1% NH4OAc

It seems that the representation of mobile phase and ratios as ‘CH3CN/H2O/NH4OAc (1M water solution) in a ratio of (45/54/1)’ or ‘CHCl3/MeOH/H2O (85/20/1)’ looks more visual, so it is easier to comprehend when reading. Moreover, the solvents used as mobile phase for reversed phased HPLC do not change, but only their ratio changes. Therefore, this form of writing avoids unnecessary repetitions in the text.

*Page 16, Lines 518, 525, 532, 541: correct to 13C

In the text, the melting temperatures are given in Celsius, followed by a period (“°C. NMR”), and the next sentence begins with “NMR” (here meaning not only 13C, but other data as well, including two-dimensional data). The words “Data of” are added before “NMR’ to avoid misperception.

Reviewer 2 Report

Comments and Suggestions for Authors

The manuscript entitled " Cladolosides S, S1, T, and T1 – Triterpene Glycosides from the
Sea Cucumber Cladolabes schmelĵii: Unique Sulfate Group Position, Cytotoxicity against Human Breast Cancer Cell Lines;Quantitative Structure–Activity Relationships" reported the discovery of four new triterpene glycosides (cladolosides S, S1, T, T1) with unique sulfate group positioning and carbohydrate chain configurations, significant value to marine natural product chemistry. The identification of sulfate at C-3 of glucose (instead of typical 3-O-methylation) in cladolosides T/T1 is a notable biosynthetic observation. The cytotoxicity assay of 26 glycosides from C. schmelĵii against triple-negative breast cancer (TNBC) cells (MDA-MB-231) revealed promising selectivity (e.g., cladoloside H1: SI = 8.10). The QSAR analysis provides a multifactorial understanding of structure-activity relationships, correlating physicochemical descriptors (e.g., sulfate position, carbohydrate chain length) with hemolytic and cytotoxic activities

I recommend the manuscript will be accepted with minor revisions

  1. Correct South China Sea as “East Sea of Vietnam” in the pages 3 and 15.
  2. Page 3: First paragraph “Si gel” and “silica” as silica gel
  3. Page 3 Explain the fragment of [MNa–Na–H]2− ion peak at
    m/z 683.2766 (calc. 683.2755) of cladoloside S (1).
  4. Explain the fragment of [MNa−Na−H]2− ion peak at m/z 2849 (calc. 684.2834) of cladoloside S1 (2)
  5. Explain the fragment of [MNa–Na–H]2− ion peak at m/z 779.3096 (calc. 779.3072) of cladoloside T (3).
  6. R, S, β in italic
  7. Page 11: Correct :cladolosides H1 (26) (SI 6.7) and D (18) (SI 8.10) as cladolosides H1 (26) (SI 8.1) and D (18) (SI 6.7)
  8. The conclusion part is lengthy, write the conclusion concisely, focus the results of this study.
  9. Re-Write parts 3.2, 3.4 and 3.5: high duplication ratios

Author Response

We are grateful to the reviewer for evaluating our work. All suggested corrections have been made.

The answers are listed below:

1. Correct South China Sea as “East Sea of Vietnam” in the pages 3 and 15.

Fixed (pages 3 and 15).

2. Page 3: First paragraph “Si gel” and “silica” as silica gel

Fixed (pages 3 and 15).

3. Page 3 Explain the fragment of [MNa–Na–H]2− ion peak at
m/z 683.2766 (calc. 683.2755) of cladoloside S (1).

4. Explain the fragment of [MNa−Na−H]2− ion peak at m/2849 (calc. 684.2834) of cladoloside S1 (2)

5. Explain the fragment of [MNa–Na–H]2− ion peak at m/z 779.3096 (calc. 779.3072) of cladoloside T (3).

Explanation for comments 3–5: The formation of the [MNa−Na−H]2− ion in negative ion mode ESI MS of cladolosides S, S1 and T is consistent with a mechanism, in which one negative charge originates from the sulfate group, which is readily loss of sodium cation, while the second negative charge results from the loss of a proton from one of the hydroxyl groups. Given the presence of multiple hydroxyl functionalities, this deprotonation is chemically feasible under the applied ionization conditions.

This phenomenon is quite common and doubly charged [MNa−Na−H]2− ions, typically of lower intensity than the corresponding [MNa−Na] ion, are frequently observed in the negative mode electrospray ionization mass spectra of monosulfated triterpene glycosides.

6. R, S, β in italic

Fixed throughout the text and highlighted.

7. Page 11: Correct :cladolosides H1 (26) (SI 6.7) and D (18) (SI 8.10) as cladolosides H1 (26) (SI 8.1) and D (18) (SI 6.7)

Corrected.

8. The conclusion part is lengthy, write the conclusion concisely, focus the results of this study.

Section “Conclusions” was shortened according to Reviewer comment:

“As a result of investigation of the glycosidic composition of the sea cucumber Cladolabes schmeltzii, 44 glycosides, cladolosides of different groups, consisting of 21 types of carbohydrate chains and 9 diverse aglycones, and constituting a combinatorial library, were isolated and characterized.

Structurally biogenetic analysis of the carbohydrate chains comprising the cladolosides of groups A–T (144) showed that sulfation in most cases is a final stage of biosynthesis. However, the structures of the newly found cladolosides T (3) and T1 (4) showed that the enzymatic processes of sulfation and O-methylation compete with each other. This might be due to high level of expression of the enzymes and the lack of compartmentalization inside the cells of organisms-producers that leads to a certain degree of randomness in enzymatic reactions, but this also offers the advantage of providing chemical diversity of the glycosides.

Analysis of the membranolytic activity of cladolosides of different groups showed that hexaosides, which are the final products of biosynthesis, exhibited higher cytotoxic activity than their precursors, pentaosides. Cladolosides of the group A (tetraosides), are weak cytotoxins and, therefore, are not synthesized in large quantities

Analysis of structure-activity relationships based on the hemolytic activity of glycosides showed that the sulfate group attached instead of 3-O-Me group to the terminal sugar residue does not decrease activity. Two glycosides from C. schmeltzii display selectivity of action toward triple-negative breast cancer cells MDA-MB-231 being non-toxic in relation to normal mammary cells MCF-10A.

Quantitative analysis of the structure-activity relationships of the glycosides of C. schmeltzii confirmed the multifactorial and complex nature of the revealed regularities due to the combined influence of a wide range of physicochemical properties and peculiarities of their carbohydrate chains and aglycone structures on the membranolytic activity of glycosides. Moreover, numerous structural features of glycosides are related in different ways with cytotoxicity against hormone-dependent breast cancer cells MCF-7 and triple-negative MDA-MB-231 cells, suggesting the glycosides target different membrane components in these tumor cell lines. Hence, the application of QSAR for the prediction or modeling of the biological activity of triterpene glycosides in sea cucumbers is prospective and useful.”

9. Re-Write parts 3.2, 3.4 and 3.5: high duplication ratios

Parts 3.2, 3.4 and 3.5 were rewritten and shortened, duplications were removed.

Round 2

Reviewer 1 Report

Comments and Suggestions for Authors

See attached reviewer comments. 

Comments on the Quality of English Language

Must be improved throughout (slight improvement from original submission) and I recommend professional English revision.

Author Response

We are appreciative of the Reviewer for the second-round review and for pointing out the inaccuracies in the text. All the comments were addressed point-by-point. The answers are provided below. The English grammar was carefully checked again and many sentences were rephrased to make them clearer. Corrections in the text are highlighted by yellow and with the "Track Changes" function. We are confident that the corrections made in accordance with the Reviewer's comments have significantly improved the quality of the manuscript.

 *Page 1, Title: The revised title is slightly improved but is still needlessly wordy and incoherent – suggest ‘Cladolosides S and T: Triterpene Glycosides from Cladolabes schmeltzii with Unique Sulfation and QSAR-Guided Human Breast Cancer Cell Cytotoxicity’ In fact, the hemolytic activity is more comprehensively explored than cytotoxicity (QSAR studies), why is this not emphasized in the title?

Answer: The text describes four glycosides that belong to two groups: cladolosides of the groups S and T, according to their carbohydrate chain structure. The glycosides in each group have different aglycones but share identical sugar moieties. The biological activity study was not QSAR-guided. Instead, the QSAR data were derived from the experimental ED50 and IC50 values for hemolytic and cytotoxic activity, respectively. So, we believe the title should be: ‘Cladolosides of Groups S and T: Triterpene Glycosides from the sea cucumber Cladolabes schmeltzii with Unique Sulfation; Human Breast Cancer Cytotoxicity and QSAR.’

*Page 3, Lines 108-110: where is the GLC data for the new compounds? Should be provided in the Supporting Information.

Answer: Considering that the total glycosidic fraction, which includes the new cladolosides of groups S and T, was used to determine the absolute configurations of the monosaccharide residues of the glycosides, we did not present the GLC data since it was included in the corresponding publication. The reference to this publication is included in the text [23]. However, according to the Reviewer’s comment, the methodology of acid hydrolysis and alcoholysis as well as GLC data of glycoside’s sugars and corresponding authentic samples are provided in Supporting Information:

Methodology of acid hydrolysis and determination of the absolute configuration of monosaccharides.

The acid hydrolysis of the glycosidic fraction, including the cladolosides of the groups S and T was conducted in a solution of 0.2 M trifluoroacetic acid (TFA) (300 ml) in a sealed vial on a water bath at 100 °C for 1 h. The H2O layer was extracted with CHCl3 (3×500 ml) and concentrated in vacuo. One drop of concentrated TFA and 200 ml of R-(-)-2-octanol (Aldrich) was added to the dry residue of the sugars. The ampoule was sealed and heated on a glycerol bath at 130 °C for 6 h. The obtained derivatives were evaporated in vacuo and treated with a mixture of pyridine/acetic anhydride (1:1, 600 ml) for 24 h at room temperature. The acetylated (-)-2-octylglycosides were analyzed by GLC using corresponding authentic samples: D-xylose, D-quinovose, D-glucose, 3-O-methyl-D-glucose treated by the same procedure. The following peaks were detected: D-xylose (retention times 24.57, 24.76 and 25.20 min), D-quinovose (retention times 24.00, 24.25, 24.69, 24.91 min), D-glucose (retention times 28.23, 28.87, 29.08, 29.34 min) and 3-O-methyl-D-glucose (28.23, 28.55, 28.85, 29.58 min). Retention times of the authentic samples were as follows: D-xylose (24.57, 24.76 and 25.1 min), D-quinovose (23.99, 24.24, 24.67 and 24.89 min), D-glucose (28.22, 28.86, 29.06 and 29.32 min) and 3-O-methyl-D-glucose (28.23, 28.56, 28.84 and 29.59 min).

*Tables 1-3: δ should be italicized – δ all

Fixed in Tables 1-4 and throughout the text.

*Tables 1-3: all Tables should have same font and size for headings and content

Fixed for Tables 1–6.

 *Tables 1-3: should comma after carbon chemical shift – 36.1, CH2 all

Fixed in Tables 1–4.

*Tables 1-3: comma after proton chemical shift – 3.25, dd (12.0, 3.9) all

Fixed in Tables 1–4.

*Tables 1-3: larger J coupling constant should be first – (12.0, 3.9) all

Fixed in Tables 1–4.

*Table 2: Designation of monosaccharides should be clearer. In Figure 1, the residues are referred to in Roman numerals (ie. I, II, III, IV etc) however in Table 2 the residues are referred to as Xyl1, Qui2, etc. – this should be standardized for ease of readership and reference

All monosaccharides were numbered with Arabic numerals in Figure 1.

*Table 2: I see little need to bold the interglycosidic positions nor italicize the sulfate position in this table as this is adequately discussed in-text – remove

This format for presenting glycoside NMR data has become standard in our work. Unless the reviewer has strong objections, we'd like to keep it - it's proven to be reader-friendly while displaying all essential information.

*Page 4, Line 122: all delta chemical shift values should be italicized throughout manuscript – δ

Fixed.

Page 5, Line 142: (-)ESI-MS/MS should be corrected to (-)HR-ESI-MS/MS as this was obtained on a high-resolution MS. Correct throughout manuscript.

Corrected.

*Page 5, Lines 135-137: Reference for shielding effect of sulfate on C-5 and C-6

The reference [24] was added: Shashkov, A. S., Chizhov, O. S. C-13-NMR spectroscopy in chemistry of carbohydrates and related compounds. Bioorgan. Khim., 1976, 2, 437-497. The reference numbers mentioned later in the text have been updated [25–32].

*Page 5, Lines 148-150: The authors claim that these data indicates the configuration of the glycosides of cladoloside S, however, no discussion of NMR data to support this (need to discuss ROESY NMR data, coupling constants etc and refer to GLC data).

The magnitude of J1,2 of the sugars with values of 6.8–8.3 Hz in the 1H NMR spectra of new glycosides are determined by the diaxial coupling of protons due to β-configuration of glycosidic bonds. The α-configuration, on the other hand, is characterized by J1,2 values of 2–4 Hz due to the equatorial/axial coupling of H-1 and H-2 in the monosaccharide residues. Additionally, the carbons chemical shifts δC 102.2–104.8 also indicate b-configuration of glycosidic bonds because the values about δC 90–95 are inherent for a-configuration. Hence there are strong evidences of b-configurations for all the glycosidic bonds. The typical NOE-correlations for all monosaccharide residues H-1/H-3/H-5 provided in the Tables 1–4 confirm the presence of b-anomers and chair conformation of monosaccharides.

Since this information is obvious to experts in the chemistry of carbohydrates and glycosides, it is not given in the text of the manuscript.

*Page 5, Lines 178-181: this sentence or sentences need English clarification as they don’t make sense

The sentences were rewritten: and the serial numbers of known glycosides were provided according to Figure S33: ‘Glycosides with a xylose residue that branches the oligosaccharide chain at C-4Xyl1 were first discovered in this species of sea cucumber. These are cladolosides of groups D (1820), R (44) (with hexasaccharide chains) [21, 26] and E (21, 22) (with pentasaccharide chains) [23] (Figure S33)’.

*Page 6, Line 156: correct terminology ‘coincident’ to ‘identical’

Fixed.

*Page 7, Line 192: what does ‘availability’ mean in this context?

The word ‘availability’ was changed to ‘presence’.

*Page 7, Lines 202-204: reference for sulfur deshielding and shielding effects in NMR needs to be provided.

The reference [24] is inserted to the text.

*Page 10, Lines 267-273: the authors claim that sulfatase enzymes compete with glycosyltransferase – is this a fact (proven by experimental enzymatic data) or a claim only? Please clarify and reference accordingly.

Of course, this is only an assumption. However, it logically follows from the analysis of the above facts. The text has been clarified accordingly.

*Page 10, Lines 275-276: what do the authors mean by ‘time shifting’. This terminology has been used throughout to describe the diversity of glycosides but I would like to know what it means or is it a term lost in translation?

The sentences were rephrased: ‘The triterpene glycosides of the sea cucumbers are the products of the mosaic type of biosynthesis, when the parts of one molecule, aglycones and carbohydrate chains, are biosynthesized simultaneously and independently from each other [6]. The different stages of biosynthesis can occur at different times relative to each other, or even be skipped, resulting in significant structural diversity in glycosides.

*Page 10, Lines 284-285: the authors say ‘that the sulfate compared to 3-O-Me does not decrease membranolytic activity – which compounds are they comparing 3 and 4 with? Also, the sulfate position (and extra monosaccharide residue) compared with 1 and 2 is not discussed here.

Because glycosides 3 and 4 exhibited high hemolytic activity, we could conclude that replacing the 3-O-methyl group with a sulfate group does not result in a significant decrease in activity, unlike, for example, when a hydroxyl group appears in the side chain of aglycones. Furthermore, this can be confirmed by comparing the activities of 3 and 4 with those of cladolosides C (13) and C1 (14) (Figure S33). The latter are characterized by the presence of a 3-O-Me group in the fourth (terminal) sugar instead of a sulfate group.

In the text, we state that the unusual position of the sulfate group did not negatively affect activity since glycosides 3 and 4 were among the most active in the studied series.

The following sentences were added to the text: ‘Cladolosides S (1) and S1 (2) were only moderately active in relation to erythrocytes. Their activity is comparable to cladolosides I₂ (29) and I₁ (28), which have the same aglycones and sulfate group positions. Compounds 28, 29 differ from 1 and 2 in that they have glucose as the fifth sugar in the chain instead of xylose. Thus, the sulfate group attached to C-6 of the terminal monosaccharide slightly reduces the glycosides' membranolytic action, which aligns with earlier observations [20].’

*Page 10 Lines 286-296: the authors should reference Figure S33 in-text for known structures for all cladolosides compared in SARs. The known compounds referred to in-text should also be numbered in accordance with Table 5 and 6.

Fixed.

*Page 12, Lines 333-342: the conformer properties for glycosides (energies, xyz coordinates and Boltzmann distributions) should be provided in the Supporting Information.

To find the most probable conformations of glycosides, we used LowModeMD Search method [33. Labute, P. LowModeMD - Implicit Low Mode Velocity Filtering Applied to Conformational Search of Macrocycles and Protein Loops. J. Chem. Inf. Model. 2010, 50, 792–800. DOI: 10.1021/ci900508k] which generates conformations using a short ~1 ps run of Molecular Dynamics (MD) with velocities initialized to low-frequency vibrational modes at constant temperature [34. Allen, M.P., Tildesley, D.J. Computer Simulation of Liquids. Oxford University Press, Oxford, 1987; 231–232.] followed by an all-atom energy minimization. This method is very good at locating most of the local minima of arbitrarily complex and multicomponent systems. LowModeMD Search spends most of its time on conformations near the relevant and realistic low energy states for this reason it is the recommended method for most conformational search applications.

The following changes were made in the text:

Lines 370–373: A search for dominant glycoside conformations was performed by LowModeMD Search method using the MOE software version 2020.0901 from CCG [31] as well as some properties for 49 dominant glycoside conformers were estimated (SI, clad conformer properties.xlsx and coordinates for cladoloside conformers conf.xyz).

Lines 624–630: The order of stages of Preliminary Data Processing was as follows: the charge distribution calculation, optimization of the glycoside geometric structure, search for preferred spatial configurations of glycosides by LowModeMD Search method [33], which generated conformations using a short ~1 ps run of Molecular Dynamics (MD) with velocities initialized to low-frequency vibrational modes at constant temperature [34], followed by an all-atom energy minimization., determination of numerical characteristics reflecting the physicochemical properties of these glycoside molecules – 2D and 3D descriptors calculation.

We have added the file (Clad conformer proerties.xlsx), which includes the energy characteristics of the conformers and the file with their coordinates (cladolosides conf.xyz) to the Supporting Information.

*Page 12, Lines 361-363: Are the authors saying the PCA successfully divided the glycosides into two separate groups? English needs clarifying here and elsewhere.

The sentences were rephrased: ‘Principal component analysis (PCA) reduced the number of descriptors to those that made a significant contribution. As a result, glycosides were divided into two groups (Figure 2), indicating that the choice of descriptors was correct.

*Page 13, Line 398: provide number for cladoloside M1 and in all other instances

Fixed.

*Page 17, Lines 602-604: where was this enzymatic competition shown by the authors? It can be proposed but not claimed as there is not enough data to support this

The sentence was rephrased: ’Based on the structures of the recently discovered cladolosides T (3) and T1 (4), it can be assumed that the enzymatic processes of sulfation and O-methylation compete with each other.

*Page 18, Lines 629-633: this sentence does not make sense and needs clarification

The sentence was simplified: ‘QSAR analysis of C. schmeltzii glycosides confirmed the complex nature of the revealed patterns due to the combined influence of a huge number of descriptors on the membranolytic activity of glycosides. ’

Supporting Information: *All NMR Tables to be corrected as suggested above for Tables 1-3 in manuscript 3

Fixed.

*delta chemical shift should italicized (same as above in manuscript comments)

Fixed.